# From Food Waste to Sustainable Agriculture: Nutritive Value of Potato By-Product in Total Mixed Ration for Angus Bulls

**DOI:** 10.3390/foods13172771

**Published:** 2024-08-30

**Authors:** Changxiao Shi, Yingqi Li, Huili Wang, Siyu Zhang, Jiajie Deng, Muhammad Aziz-ur-Rahman, Yafang Cui, Lianqiang Lu, Wenxi Zhao, Xinjun Qiu, Yang He, Binghai Cao, Waseem Abbas, Faisal Ramzan, Xiufang Ren, Huawei Su

**Affiliations:** 1State Key Laboratory of Animal Nutrition and Feeding, College of Animal Science and Technology, China Agricultural University, Beijing 100193, China; scx1107@cau.edu.cn (C.S.); liyingqi1230@163.com (Y.L.); s20223040747@cau.edu.cn (H.W.); zsycaucast@163.com (S.Z.); 18811560713@163.com (J.D.); 15506016835@163.com (Y.C.); lulianqiang1996@163.com (L.L.); 2022303090310@cau.edu.cn (W.Z.); qiuxinjun@hainanu.edu.cn (X.Q.); heycau@163.com (Y.H.); caobh@cau.edu.cn (B.C.); 2Institute of Animal and Dairy Sciences, University of Agriculture Faisalabad, Faisalabad 38040, Pakistan; drazizurrahman@uaf.edu.pk (M.A.-u.-R.); waseem.abbas@uaf.edu.pk (W.A.); drfaisal.ramzan@uaf.edu.pk (F.R.); 3Shangdu County Animal Husbandry Service Center, Shangdu County, Ulanchap 013450, China; scx15646702767@163.com

**Keywords:** potato by-product, beef cattle, daily weight gain, rumen microbiota

## Abstract

Raw potato fries are a type of potato by-product (PBP), and they have great potential as a partial replacement of grain in animal feeds to improve the environmental sustainability of food production. This study aimed to investigate the effects of replacing corn with different levels of PBP (0%, 12.84%, 25.65%, and 38.44%) in the total mixed ration (TMR) of Angus bull. Sixty 16-month-old Angus bulls (548.5 ± 15.0 kg, mean ± SD) were randomly assigned to four treatments. The results indicated that with the increase in the substitution amount of PBP, the body weight decreased significantly. The dry matter apparent digestibility and starch apparent digestibility linearly decreased as PBP replacement increased. The feed ingredient composition in the TMR varied, leading to a corresponding change in the rumen microbiota, especially in cellulolytic bacteria and amylolytic bacteria. The abundance of *Succiniclasticum* in the 12.84% PBP and 38.44% PBP diets was significantly higher than that in the 0% PBP and 25.65% PBP diets. The abundance of *Ruminococcus* linearly increased. In conclusion, using PBP to replace corn for beef cattle had no negative impact on rumen fermentation, and the decrease in apparent digestibility explained the change in growth performance. Its application in practical production is highly cost-effective and a strategy to reduce food waste.

## 1. Introduction

Non-forage carbohydrates are a good source of dietary energy and are considered very palatable for beef cattle [1]. Grain contributes a major portion to the diet of fattening beef cattle; however, feeding a grains-rich diet to ruminants usually leads to health-related issues, particularly subacute ruminal acidosis [2]. Another problem with high grain feeding in ruminants is its competition with human food, as most cereal grains are human-edible ingredients. Thus, cereal grain reductions in ruminant feeding have become a production priority. Ke et al. (2021) reported that the agricultural industry produces 998 million tonnes of by-products globally that need to be disposed of, and these by-products are normally burned or used as compost, which leads to possible environmental problems [3]. Feed grain replacement can bring an array of cascading benefits such as land and fertilizer spared, feed costs reduced, and pollution avoided [4]. Thus, the availability of by-products from various food industries could be potential alternative energy ingredients for animal nutrition [5,6]. 

Potatoes, together with rice, wheat, and maize, make up the four crops that supply 50% of the world’s food energy needs [7]. Although potato is classified as a high glycemic index (56–125) food due to its rich carbohydrate content (9.1–22.6 g/100 g fresh weight) [8,9], it also contains a lot of healthful nutrients such as protein (1–4.2%), resistant starch (10 g/100 g fresh weight), and phenolic compounds [8,10]. In recent years, worldwide potato production has fluctuated around 360 million tonnes [11]. With the development of the potato industry, 81 million tonnes of potato by-products (PBPs) were produced [12]. In some countries, such as Mexico, the proportion of potato waste in the potato product processing chain has reached as high as 37.11% [13]. Recently, a new range of feed products derived from potato by-products (PBPs) has emerged, driven by the trend in developed countries towards the consumption of ready-prepared meals and fast food items like French fries. The PBP waste is generated from the production of commercial raw fries, which consists of raw potato fries that do not meet size standards after steam peeling, cutting, and sieving processes that could be potential feed ingredients for ruminants. It has been reported that PBP has great potential for use as an animal feed, and PBP could be an alternative to cereal grains because of its low price and superior nutritional value with high energy for beef cattle and pigs [14,15,16]. Most of the previous studies that examined the utilization of PBP for animal diets were in the form of fresh, dried, ensiled, or steamed [16]. It has been reported that normally, PBP contains dry matter (DM) ranging from 10% to 30%, starch 34.8% to 56.3% on a DM basis, and crude protein (CP) 3.8% to 17.2% on a DM basis [17]. Furthermore, compared to other plant proteins, potato protein is considered a valuable non-animal protein due to its high essential amino acid concentration [18]. Neutral detergent fiber (NDF) and acid detergent fiber (ADF) concentrations of potato by-product are at 35.3% to 46.2% and 15.3% to 34.2% of DM, respectively, which varies depending on the proportion of different fractions of potatoes in the final feedstuff [16].

However, the processing residues contain substances harmful to animals, such as glycolkaloids and nitrates [19]. Therefore, farm owners prefer to use by-products such as peeled raw fries and fried fries to avoid the impact of adverse factors in the beef cattle industry [20]. An additional benefit of using PBP from raw fries production waste could be related to chemical changes during baking, grinding, and heating. Manufacturing processes increase ruminal starch degradation, which could affect feed intake as well as rumen metabolic health [21,22]. Most importantly, a possible increase in starch digestibility by processing, grinding, and baking might cause a rapid decrease in ruminal pH and, hence, subacute rumen acidosis conditions [23]. 

The use of PBP in livestock diets has been examined for lactating dairy cows [24], beef cattle [17], and sheep [25]. It has been reported that a 10% replacement of corn grain with PBP in the feed of beef cattle increases dry matter intake (DMI) and weight gain [26]. Another study by Charmley et al. (2006) observed that PBP could replace maize and barley grain by up to 40% without any negative effects on animal growth, carcass trait, or meat quality [27]. Radunz et al. (2003) concluded that the addition of no more than 30% PBP to beef cattle diets showed no significant effect on beef Warner–Bratzler shear force, juiciness, and flavor intensity [28]. However, another study reported that replacing barley in beef cattle diets with PBP decreased carcass weight but resulted in overall economic benefits [29]. The production of volatile fatty acids (VFA) from rumen fermentation is an important component of energy utilization in ruminants [30]. VFA composition is altered by different feed ingredients [31]. Different types of potato waste have different effects on rumen fermentation characteristics in beef cattle [28,32]. The fundamental reason for the variation in VFA is the alteration of rumen microorganisms [33]. Rumen microbiota is crucial for the digestion and absorption of nutrients in the rumen, which is ultimately closely related to animal growth and health conditions [34,35]. Hence, microbial population data can inherently improve the understanding of host animal growth performance dynamics and can eventually allow livestock producers to improve animal production-related decisions [36]. The effect of PBP on the rumen microbial population of beef cattle has not been reported yet. In order to better investigate the effects of PBP on rumen fermentation after feeding to beef cattle, it is necessary to explore the relevant microbiota that lead to changes in rumen fermentation parameters. This will reveal the reasons for the changes in beef cattle performance after feeding PBP. 

Based on the current reality of reducing food waste and rising feed costs, this study aims to evaluate the effects of the graded substitution of cereal grains by PBP on the growth performance, apparent total tract digestibility, and ruminal fermentation parameters of beef cattle under the same DMI. To explain the differences in animal growth performance, further research was conducted on the rumen microbiota. We hypothesized that substituting corn with PBP up to a 75% threshold in the diet of Angus bulls will not result in detrimental effects on their growth performance or rumen fermentation.

## 2. Materials and Methods

### 2.1. Animal Ethics

All procedures involving animal handling and treatment were approved by the China Agricultural University Laboratory Animal Welfare and Animal Experimental Ethical Inspection Committee (Permit No. AW 72303202-1-1). Methods were reported in the manuscript following the recommendations in the ARRIVE guidelines.

### 2.2. Description of Potato By-Product Sources

The PBP used in this experiment was procured from a local feed company LambWeston (Ulanqab, China) Co., Ltd., before the frying process of a batch. Collected PBP was wasted from the production of commercial raw fries and comprised of raw potato fries with substandard size after steam peeling, cutting, and sieving. Prior to delivery, PBP was stored in a processing plant warehouse at 0 to 4 °C, and PBP was transported 2 to 3 times per week. Fresh PBP was stored in a separate feed storage area (no more than 2 days) on the farm. The sampling frequency and nutrient analysis were consistent with the TMR. The chemical analysis of the PBP used in the current experiment, as well as that of corn grains used to substitute them, are shown in Table 1.

### 2.3. Animals, Diets, and Experimental Design

Sixty 16-month-old Angus bulls (548.5 ± 15.0 kg, mean ± SD) were housed in separate outdoor pens based on a randomized block design. Animals were blocked based on body weight (BW). Four treatment (15 bulls per group) diets were designed according to the NASEM (2016) under the reference of the fattening cattle production, lasting 97 days, including the first 7 days as an adaptation period, with the method of limiting the diet intake to maintain no orts [37]. All animals had equal DMI during the experimental period (Appendix A for DMI and metabolizable energy intake changes). Adjustments were made daily according to the actual feed intake of the experimental cattle. Experimental diets were prepared by using PBP to replace different proportions of corn (filtered through a 3 mm local commercial grain sieve) in the TMR. Animals were fed TMR containing different levels of ground corn or PBP, commercial concentrate, corn silage, and wheat straw. On a DM basis, the control group was fed 0% PBP but 50.98% ground corn grains in TMR. Whereas the other 3 experimental groups were fed either 12.84% PBP plus 38.20% ground corn grains in TMR (12.84% PBP diet), 25.65% PBP plus 25.43% ground corn grains (25.65% PBP diet) in TMR, or 38.44% PBP plus 12.70% ground corn (38.44% PBP diet) in TMR. A detailed list of ingredients and chemical composition of the diets from laboratory analysis is shown in Table 2. The TMR was supplied to bulls at 09:00 (GMT + 8) and 16:00 (GMT + 8). Fresh drinking water was freely available throughout the study. 

### 2.4. Sampling, Data Collection, and Chemical Analysis

Animals were weighed 4 times (d 0, 31, 61, and 91) during the experimental period under fasted conditions before morning feeding and averaged over two consecutive weighings using a pre-calibrated professional livestock weighing scale. The average daily gain (ADG) was calculated as body weight gain (BWG) divided by the number of experimental days and feed conversion ratio (FCR) as DMI/daily BWG. The data of ADG and FCR were calculated four times (0 to 30 d, 31 to 60 d, 61 to 90 d, and 0 to 90 d). Economic benefits are based on the actual situation (Appendix A). Samples of fresh TMR from Table 2 and other raw materials of d 0, 31, 61, 86, 87, 88, 89, and 90 during the experimental period of 90 days were collected for nutrient analysis (Table 2). The fecal samples were collected from the rectum of each animal in the last 3 d of the experiment (88 to 90 d). Tartaric acid (10%, *v*/*v*) at a ratio of 1:4 (g/mL), the mass of the fresh fecal samples was added and mixed well for nitrogen fixation. The sampled feces were then stored at −20 °C for later analysis. In the TMR, raw materials and fecal samples, NDF, ADF, and acid-insoluble ash were determined by Van Soest (1981) [38], ash (AOAC Official Method 942.05), DM (AOAC Official Method 934.01), EE (AOAC Official Method 920.39), and CP (AOAC Official Method 984.13) were analyzed according to the AOAC (2002) [39]. Starch was analyzed according to Tang (2012) [40]. The apparent total tract digestibility was calculated using acid-insoluble ash as an internal marker, as previously described by Keulen et al. (1977) using the following formula [41]:Apparent total tract digestibility (%) = 100 − (Ad × Nf)/(Af × Nd) × 100
where Nd is the content of a given nutrient in the diet (%), Nf is the content of the same nutrient in the feces (%), Ad is the content of AIA in the diet (%), and Af is the content of AIA in the feces (%).

Rumen fluid sample was collected on the last day (d 90) of the experiment from 32 bulls (eight cattle per group), 3 h after morning feeding via a stomach tube. In order to exclude the influence of saliva on rumen fluid pH, the first 200 mL of rumen fluid sample was discarded to minimize contamination from the saliva. After that, approximately 200 mL rumen fluid was collected, and fluid pH was determined immediately (testo 205, Lenzkirch, Germany); then, the sample was filtered through 4 layers of medical gauze. Filtrate used for measurement of ammonia nitrogen (10 mL) and VFA (10 mL) was stored at −80 °C and premixed with 2.5 mL metaphosphoric acid (250 g/L). Ammonia nitrogen was measured following the method described by According to Weatherburn (1967) [42], and VFA was determined by gas chromatography (GC-2014 Shimadzu Corporation, Kyoto, Japan) with 2-ethyl butyric acid as an internal standard.

DNA from Angus bull rumen fluid samples was extracted using the TGuide S96 Magnetic Soil/Stool (Tiangen Biotech (Beijing, China) Co., Ltd.). The V3-V4 region of the prokaryotic small-subunit (16 S) rRNA gene was amplified with primers 338F (5′-ACTCCTACGGGAGGCAGCA-3′) and 806R (5′-GGACTACHVGGGTWTCTAAT-3′) [43]. Meanwhile, we detected and analyzed the *16S* rRNA gene according to the methods Li et al. (2023) reported [44]. The bioinformatics analysis of this study was performed with the aid of the BMK Cloud (Biomarker Technologies Co., Ltd., Beijing, China).

### 2.5. Data Analysis 

The statistical power of the sample size was analyzed based on the methods of Friedman (1982) [45]. We calculated all the statistical power using G*Power software [version 3.1.9.7, https://g-power.apponic.com, accessed on 15 July 2023; Faul et al. (2007, 2009)] to obtain power (1 − β) of 0.80 and a type-I error probability (α) of 0.05. Each animal was referred to as an experimental unit in all analyses [46,47]. Growth performance results were made with Graph Pad Prism software 8.0.2.

The experimental data were sorted by Excel 2019, and SPSS 25.0 was used for fitting a general linear model (GLM) with the following equation: *Y_ij_* = *μ* + *τ**_i_* + *ϵ_ij_*
where *Y_ij_* is the dependent variable, *μ* is the common effect of the whole experiment, *τ_i_* represents the treatment group dietary effect, and *ϵ_ij_* represents the random error present in the *j_th_* observation of the *i_th_* diet. 

The α diversity was calculated and displayed by the QIIME2 and R (version 4.0.3, ref), respectively. The β diversity was determined to evaluate the degree of similarity of microbial communities from different samples using QIIME.

Multiple comparisons were determined by Duncan’s multiple comparison tests to identify the significance of treatment differences. Significance was declared at *p* ≤ 0.05. Trends were recognized at 0.05 ˂ *p* ≤ 0.10.

## 3. Results

### 3.1. Description of Results

#### 3.1.1. Growth Performance

Figure 1 and Appendix A represent the effect of PBP replacing different proportions of corn on the growth performance of Angus bulls. Figure 1A showed that the final BW of the 0% PBP and 12.84% PBP experimental diets were greater than that of the 38.44% PBP diet (*p* = 0.003). With the increase in the substitution amount of PBP and FCR was increased linearly from approximately 0 to 30 d and 0 to 90 d (*p* < 0.05) (Figure 1B,C). Similarly, the increase in the substitution amount of PBP and ADG was decreased linearly from the approximate 0 to 90 d period, while the ADG of 0 to 30 d and 61 to 90 d period also decreased linearly (0.05 < *p* ≤ 0.10).

#### 3.1.2. Apparent Total Tract Nutrient Digestibility

Table 3 represents the effect of replacing different proportions of corn with PBP in TMR on the apparent digestibility of Angus bulls. The results showed that with the increase in PBP, the digestibility of the total diet was significantly decreased (*p* < 0.001), and the starch digestibility of the 0% PBP and 12.84% PBP experimental diets were significantly higher than that of the 25.65% PBP and 38.44% PBP diets (*p* = 0.002). The results also showed that the apparent digestibility of CP, EE, NDF, and ADF was not significantly different among the groups (*p* > 0.05).

#### 3.1.3. Ruminal Fermentation

Table 4 represents the effect of replacing different proportions of corn with PBP in TMR on the rumen fermentation of Angus bulls. Rumen pH had a decreasing trend (*p* = 0.076), which showed a linearly downward trend (*p* = 0.078); however, the other ruminal fermentation indexes were not significantly affected by the experimental treatments (*p* > 0.05).

#### 3.1.4. *16S* rRNA Gene Analysis of Bacterial Communities

Using next-generation sequencing technology, 2,546,495 reads across all samples, with an average of 79,401 per sample, were acquired. As shown in Appendix A, it can be seen from the dilution curve that as the amount of sequencing increases, the curve tends to be flat, indicating that sufficient sequencing depth has been achieved. The Good’s coverage of each sample is higher than 99.99%, indicating that the sequencing results of this experiment can represent the real situation of the microorganisms in the samples.

#### 3.1.5. Ruminal Microbiota

The results are shown in Figure 2A. The ACE (*p* = 0.077) and Chao1 (*p* = 0.077) indices tended to increase as the proportion of PBP in the diet increased. The ACE index and Chao1 index of the 38.44% PBP diet were significantly higher than the 12.84% PBP experimental diets. The Shannon index in the 38.44% PBP diet was significantly higher than that in the 0% PBP experimental diet (*p* = 0.026) and extremely significantly higher than that in the 12.84% PBP diet (*p* = 0.0033). The Simpson index of the 38.44% PBP diet was significantly higher than that of the 25.65% PBP diets (*p* = 0.031) and extremely significantly higher than that of the 12.84% PBP diet (*p* = 0.0095). There was no significant difference in each parameter among the 0% PBP, 12.84% PBP, and 25.65% PBP experimental diets. PLS-DA analysis of the rumen microbiota showed that the bacterial communities of the groups were distinctly separated, suggesting that different ratios of PBP to replace corn may alter the rumen bacterial community composition of Angus bulls (Figure 2B).

#### 3.1.6. Classification Overview

As shown in Figure 3A,B, through bacterial classification analysis, at the phylum level, there were 11 bacterial phyla with relative abundance greater than 0.1% (Appendix A). At the genus level, there were 26 genera (except *uncultured_rumen_bacterium*) with relative abundance greater than 1%. As shown in Appendix A and Figure 3B, the abundance of *Succiniclasticum* in the 12.84% PBP and 38.44% PBP diets was significantly higher than that in the 0% PBP and 25.65% PBP diets (*p* = 0.012), and the abundance of *unclassified_F082* in the 12.84% PBP, 25.65% PBP and 38.44% PBP diets was significantly higher than that in the 0% PBP diet with the increase in the replacement ratio of PBP (*p* = 0.019). The abundance of *Clostridium_sensu_stricto_1* and *Prevotellaceae_UCG_001* tended to increase (*p* ≤ 0.10), and the abundance of *Sharpea* showed a change tendency (*p* = 0.084). The abundance of *Ruminococcus* and *unclassified_Prevotellaceae* linearly increased (*p* < 0.05), and the abundance of *Fibrobacter* had a linear tendency (*p* = 0.053). The abundance of *Lachnospiraceae_NK3A20_group* quadratically increased (*p* = 0.027), and the abundance of *Prevotella_7* and *unclassified_Lachnospiraceae* had a quadratic tendency (*p* ≤ 0.10).

#### 3.1.7. Correlation Analysis

Figure 4 shows the Spearman correlation between growth performance and rumen fermentation parameters, with abundance ratios higher than 1% at the genus level. The results showed that ADG and FCR were negatively correlated with the abundance of *NK4A214_group* and *unclassified_Clostridia* (*p* < 0.05). Rumen pH was positively correlated with the abundance of four genera, including *Saccharofermentans* (*p* < 0.05, *r*^2^ = 0.5062). The concentration of acetic acid was positively correlated with the abundance of *Succinivibrionaceae_UCG_001* (*p* = 0.021, *r*^2^ = 0.4062). The concentration of propionic acid was positively correlated with the abundance of *Succinivibrio* (*p* = 0.017, *r*^2^ = 0.4172) and *Succinivibrionaceae_UCG_001* (*p* = 0.018, *r*^2^ = 0.4143). The A:P ratio was positively correlated with the abundance of three genera, including *Prevotellaceae_UCG_003* (*p* < 0.05, *r*^2^ = 0.3842), and negatively correlated with the abundance of *Succinivibrio* (*p* = 0.032, *r*^2^ = −0.3798). The total volatile fatty acid concentration was positively correlated with the abundance of *Succinivibrio* (*p* = 0.026, *r*^2^ = 0.3926) and *Succinivibrionaceae_UCG_001* (*p* = 0.006, *r*^2^ = 0.4784).

## 4. Discussion

Under almost isoenergetic and isonitrogenous TMR, this experiment demonstrated the feasibility of food waste, specifically PBP, as feed for the beef cattle industry. The feeding of all animals was carefully regulated to eliminate any potential impact of differences in DMI on their productive performance. As PBP substitution increased, starch content increased, and fiber and EE content decreased, while the other components were largely unchanged. Overall, as the proportion of PBP replacement increased, ADG decreased slightly. Due to the change in ADG and FCR in the whole period, the final BW of the Angus bulls in the 0% PBP diets was greater, but a maximum of 2.6% reduction in BW is perfectly acceptable. This may be attributed to the change in TMR composition, feed digestion level, and energy utilization efficiency. Compared with corn starch, potato starch has higher degradation in the rumen [48]. Theoretically, the bacteria in the rumen utilizing starch provide energy for beef cattle in the form of VFA, with an energy utilization rate of about 80% [49]. The undegraded starch in the rumen is absorbed and digested by the small intestine with a higher glucose digestibility and energy conversion efficiency. To be more specific, the energy utilization rate of enzymatic digestion in the small intestine is about 97% [50]. In other words, potato starch has a higher digestibility in the rumen compared to corn starch, reducing energy utilization in Angus bulls, which leads to this result. 

Feed digestibility directly affects the nutrient supply to the animal, which then influences its growth performance [37]. The difference in the TMR component is the main reason for the variation in digestibility. Sutherland et al. (2021) reported that starch apparent digestibility significantly decreased in growing heifers’ diets when starch contents were increased from 30.98% to 34.26% [51]. Similarly, Johnson et al. (2020) noted that in high-concentrate finishing beef heifers TMR, with the increase in starch content (53.1~62.0%), the apparent digestibility of starch decreased [52]. In addition, starch is an important component of non-fiber carbohydrates (NFC), and it was generally accepted that feeds with high NFC are more digestible; there was a threshold of total tract digestibility as NFC increased [53]. So, excessive starch intake may be a key factor in digestibility.

The differences in starch sources may contribute to the variations in apparent starch digestibility in Angus bulls. Bajaj et al. (2018) found that potato starch granules are oval-shaped with a diameter of 35.74 μm, while corn starch is irregularly polyhedron-shaped with a diameter of 15.08 μm [54]. Starch granules with larger sizes and smoother surfaces, along with specific supramolecular properties, tend to be more resistant to digestive enzymes. Conversely, starch granules with a rough surface and the presence of surface pores and channels can enhance enzyme diffusion and adsorption. Appropriate processing methods can potentially mitigate the negative impact of differences in starch structure on digestibility [55]. However, in this experiment, PBP was not processed compared to ground corn, which could be a potential reason for the observed difference in digestibility.

It has been speculated that the changes in digestibility are related to the moisture content of the feed. Miller et al. (2009) reported that the increasing moisture content in the TMR decreased the DMI of cows [56]. This may be attributed to poorer feed palatability and increased filling effect in the rumen. As the proportion of PBP in the diet increases, the moisture and fresh weight of the TMR increases, which leads to more adequate gastrointestinal tract fill and increased viscosity of the chow, resulting in insufficient feedstuff digestion [29]. As the level of PBP increased in the TMR, there was a significant decrease in the DM and starch apparent digestibility, which may be the most direct explanation for the decrease in growth performance in Angus bulls.

Under the conditions of this experiment, the pH value of Angus rumen fluid was in a healthy range, and rumen microorganisms can carry out normal activities and proliferation. This was consistent with the experimental results of Omer et al. (2010), who used different proportions of PBP instead of corn to feed lambs [57]. However, with the increase in the replacement amount of PBP, the rumen pH of the experimental cattle had a decreasing trend. It was presumed to be caused by the differences in starch content, composition, degradation rate, and digestion site between potato and corn. According to the in vitro experimental data, potato starch is higher than corn starch in rumen degradable fraction [21]. Hindle et al. (2005) used fistulated Holstein cattle to conduct an in vivo digestion test; it was found that the rumen digestibility of potato starch was 84.0%, while the corn starch was 75.3% [48]. At the same time, the in vitro digestibility of starch was determined by the nylon bag method, and it was found that within 45 min, potato starch degraded 40%, while corn starch degraded only 7%. For rumen digestibility, potato starch is about 5.0%/h~6.3%/h, while corn starch is 2.0%/h~6.2%/h [48,58,59], which indicates a negative correlation between the amount of rumen degradable starch and rumen pH [60]. Starch is the primary non-structural carbohydrate, and its rapid fermentation causes rapid VFA production [61]. And high-concentrate diets promote propionate production, which leads to lower PH and A:P ratio [62,63]. 

The composition and function of rumen microorganisms were closely related to the production performance of ruminants [64]. Differences in dietary nutrients also have an impact on the rumen microbiota. A study of rumen microbiota with different feed efficiencies showed that, among different groups, the abundance of *Succiniclasticum*, *Lactobacillus*, *Ruminococcus*, and *Prevotella* significantly changed, indicating that the composition and structure of the rumen microbiota affect the FCR of beef cattle [65]. In particular, *Succiniclasticum* and *Ruminococcus* are important members of the core rumen microbiota of ruminants that degrade carbohydrates in the diet to provide nutrients to the animal [66,67]. Nevertheless, there was no significant change in fiber digestibility in the current study, which was unlike previous research and could be attributed to the difference in resistant starch content causing a change in the abundance of *Ruminococcus* [68]. However, a previous study concluded that the abundance of *Ruminococcus* was negatively correlated with FCR, which is consistent with the conclusion of our study [69]. With higher PBP addition, the TMR starch content increased, which improved the abundance of *Succiniclasticum*. However, too much starch could not be attached by the bacteria and was expelled directly into the digestive tract, which led to lower starch digestibility. Consistently, the abundance of *Succiniclasticum* and *Ruminococcus* was significantly changed in the current study, which we considered as a possible result of the variation in the TMR of Angus bulls.

Potatoes contain more resistant starch than corn [70]. *Ruminococcus* can decompose resistant starch in potato to produce succinic acid, acetic acid, and formic acid [71,72]. Studies have shown that the relative abundance of *Succiniclasticum* increases significantly with increasing dietary energy levels [73]. Succinic acid was the only fermentable substrate of *Succiniclasticum*, and only propionic acid was produced [74,75]. The propionate contents are negatively related to the A:P ratio, which is similar to the findings of our study. Their abundance decreased with the increased proportion of PBP, which may be the reason for the linear decreasing trend of the A:P ratio. The increased abundance of *Succiniclasticum* in our study suggests enhanced propionate production, which could be linked to improved feed efficiency and energy utilization in the bulls. Several studies indicated a positive correlation between the abundance of *Sharpea* and the starch content in feeds, leading to a rapid production of lactic acid. This phenomenon is considered a potential factor contributing to the decrease in rumen pH [76,77]. In summary, the addition of PBP caused ruminal microbiota modification of Angus bulls, which changed key colony species and abundance, which may be an important factor affecting the growth performance of beef cattle.

The agricultural by-products that can be repurposed as animal feed are referred to as non-food-competitive feeds. These feeds offer cost-effectiveness without compromising productivity [78]. Economic results of the current study revealed that the increasing substitution of PBP with corn enhanced the profit, which was attributed to the low cost. Overall, the 38.44% PBP diet group had a 27% increase in farm profits and a 36% reduction in feeding costs during the entire trial period compared to the control group (Appendix A). In light of the nutritional and environmental losses caused by food waste, this feeding strategy offers a promising approach to turning waste into a valuable feed component for sustainable animal production [79]. Particularly in the background of food–feed competition, it was more advantageous to use this non-food-competitive feed in TMR for beef cattle. In the future, it will be a valuable research direction to explore the long-term applications of non-food-competitive feeds such as PBP at different stages in different animal breeds.

## 5. Conclusions

Under the current experimental conditions, with increased substitution of PBP with corn, growth performance showed a decreasing trend, most directly due to a lower nutrient digestibility of Angus bulls. Rumen bacteria varied with dietary changes, but rumen fermentation was not negatively affected. Considering the results of ADG and the economic benefits of feeding PBP to Angus bulls, the recommendation is to include 38.44% PBP in the fattening cattle TMR for greater profitability. This study also showed a new insight into the use of agro-forest side streams, especially food waste, while reducing nutritional and environmental losses.

## Figures and Tables

**Figure 1 foods-13-02771-f001:**
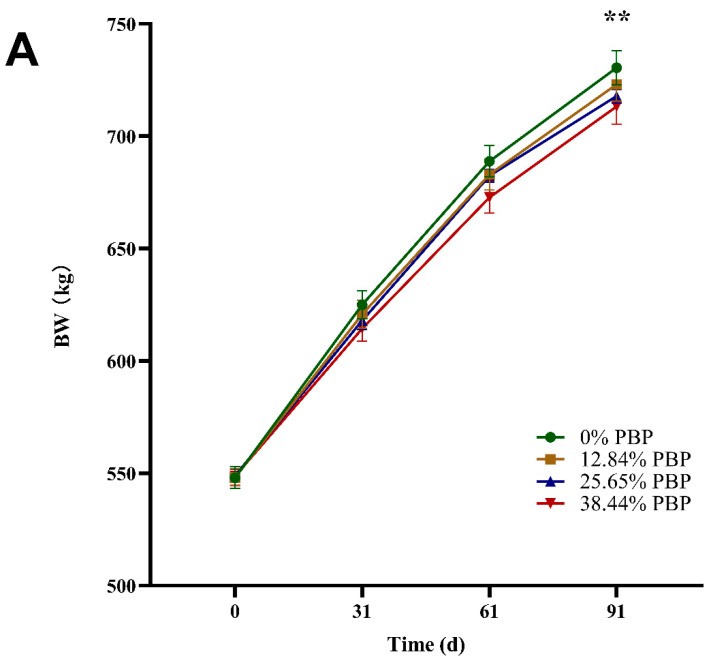
Effect of PBP replacing corn in different proportions on BW (body weight) (**A**); ADG (average daily gain) (**B**); FCR (feed conversion ratio) (**C**) of Angus bulls. Significance levels: **, *p* < 0.01.

**Figure 2 foods-13-02771-f002:**
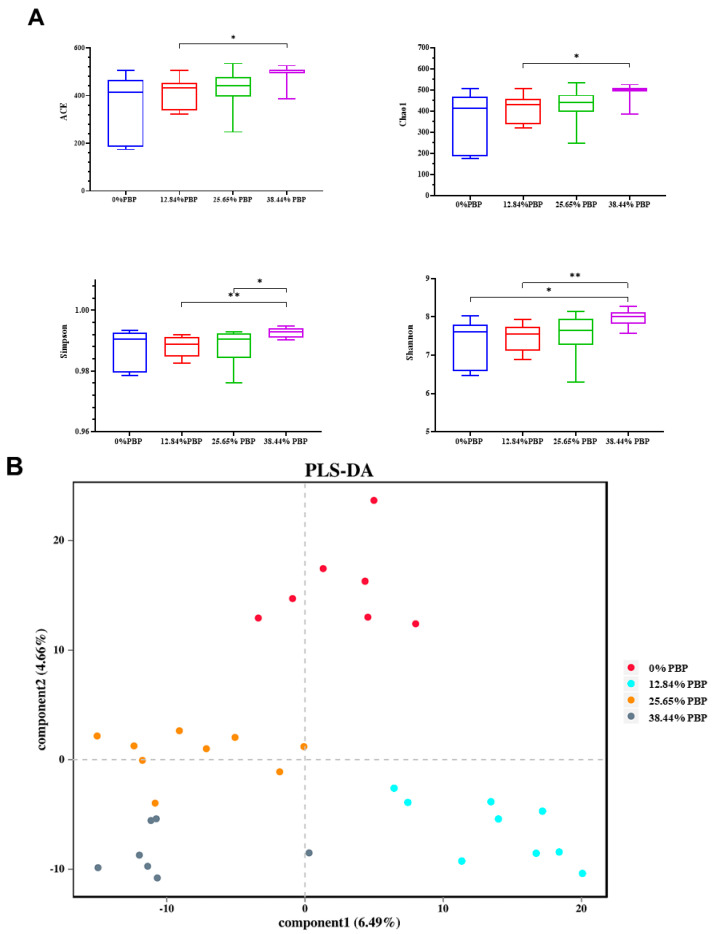
Effect of replacing different proportions of corn with PBP in TMR on the α-diversity (**A**) and β-diversity (**B**) of Angus bull rumen microbiota (data are mean ± SD, *n* ≥ 7 bulls per treatment). Significance levels: *, *p* < 0.05; **, *p* < 0.01.

**Figure 3 foods-13-02771-f003:**
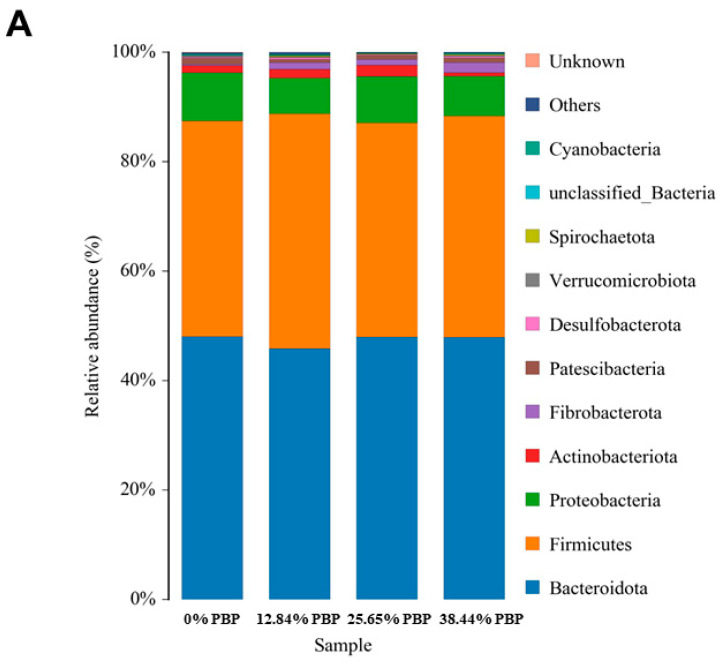
(**A**) Histogram of species distribution at the phylum level. (**B**) Histogram of species distribution at the genus level.

**Figure 4 foods-13-02771-f004:**
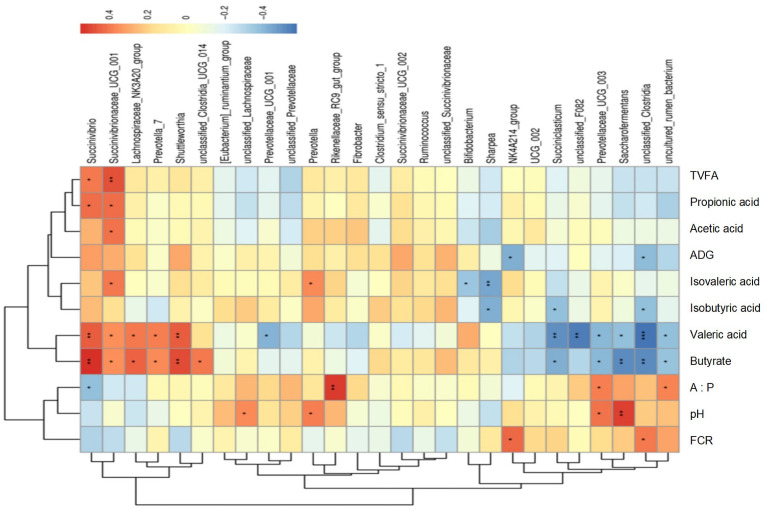
Correlation of growth performance, rumen fermentation indexes, and rumen microbiota. Strong correlations are indicated by red and blue colors, with 1 indicating a perfect positive correlation (dark red) and −1 indicating a negative correlation (dark blue), whereas weak correlations are indicated by yellow colors. Spearman test: *, *p* < 0.05; **, *p* < 0.01; ***, *p* < 0.001.

**Table 1 foods-13-02771-t001:** Comparison of the nutritional value of corn and PBP (DM basis).

Items ^1^	Ground Corn ^2^	PBP
DM, %DM	87.05 ± 0.20	21.4 ± 0.87
CP, %DM	9.35 ± 0.20	8.45 ± 0.36
EE, %DM	3.82 ± 0.17	1.44 ± 0.13
NDF, %DM	8.43 ± 0.19	5.01 ± 0.15
ADF, %DM	3.22 ± 0.11	2.53 ± 0.23
Starch, %DM	72.63 ± 0.62	79.56 ± 0.73
Ash, %DM	1.57 ± 0.08	2.02 ± 0.05
ME ^3^, MJ/kg DM	16.18	16.19

^1^ DM, dry matter; CP, crude protein; EE, ether extract; NDF, neutral detergent fiber; ADF, acid detergent fiber; chemical composition as mean ± SEM. ^2^ Ground corn is filtered through a 3 mm sieve using local commercial corn. ^3^ ME (metabolizable energy) was calculated according to the NASEM (2016) model.

**Table 2 foods-13-02771-t002:** Composition and chemical analysis of nutrient levels of experimental diets (DM basis).

Ingredients	PBP to Replace Corn in Different Proportions ^1^			
	0% PBP	12.84% PBP	25.65% PBP	38.44% PBP
Ground corn, %	50.98	38.20	25.43	12.70
PBP, %	0.00	12.84	25.65	38.44
Commercial concentrate ^2^, %	14.83	14.82	14.80	14.79
Corn Silage, %	26.42	26.39	26.36	26.34
Wheat straw, %	7.76	7.75	7.74	7.74
Chemical composition ^3^				
CP, %DM	12.69 ± 0.11	12.56 ± 0.08	12.30 ± 0.03	12.05 ± 0.09
EE, %DM	2.49 ± 0.04	2.11 ± 0.02	1.60 ± 0.05	1.19 ± 0.01
NDF, %DM	30.28 ± 0.62	29.34 ± 0.51	27.23 ± 0.42	25.39 ± 0.71
ADF, %DM	14.11 ± 0.57	13.79 ± 0.49	13.49 ± 0.17	13.24 ± 0.61
Ash, %DM	5.55 ± 0.06	5.76 ± 0.07	5.83 ± 0.10	6.02 ± 0.11
Starch, %DM	41.70 ± 0.53	43.84 ± 0.42	44.99 ± 0.53	46.16 ± 0.80
ME ^4^, MJ/kg DM	14.18	14.22	14.25	14.30

Abbreviations: ^1^ 0% PBP, control group; 12.84% PBP, 25% replacement group; 25.65% PBP, 50% replacement group; 38.44% PBP, 75% replacement group. ^2^ Commercial concentrate is shown in Appendix A. ^3^ DM, dry matter; CP, crude protein; EE, ether extract; NDF, neutral detergent fiber; ADF, acid detergent fiber; chemical composition as mean ± SEM. ^4^ ME (metabolizable energy) was calculated according to NASEM (2016) model.

**Table 3 foods-13-02771-t003:** Effect of PBP replacing corn in different proportions on apparent total tract digestibility of Angus bulls (%).

Items ^1^	PBP to Replace Corn in Different Proportions ^2^				SEM	*p*-Value		
	0% PBP	12.84% PBP	25.65% PBP	38.44% PBP		Treatment	Linear	Quadratic
DM (%)	74.81 ^a^	74.93 ^a^	73.57 ^b^	71.88 ^c^	0.253	<0.001	<0.001	<0.001
CP (%)	71.29	70.29	69.05	70.66	0.448	0.434	0.478	0.199
EE (%)	76.93	78.29	78.85	76.99	0.523	0.498	0.877	0.144
NDF (%)	62.19	62.79	62.84	59.29	0.882	0.535	0.338	0.307
ADF (%)	60.13	60.21	63.09	60.39	0.831	0.577	0.642	0.433
Starch (%)	94.06 ^a^	93.33 ^a^	90.01 ^b^	91.19 ^b^	0.514	0.002	0.001	0.102

^1^ DM, dry matter; CP, crude protein; EE, ether extract; NDF, neutral detergent fiber; ADF, acid detergent fiber. ^2^ 0% PBP, control group; 12.84% PBP, 25% replacement group; 25.65% PBP, 50% replacement group; 38.44% PBP, 75% replacement group. ^a^,^b^, and ^c^ Means within a row with different superscripts differ (*p* < 0.05).

**Table 4 foods-13-02771-t004:** Effect of PBP replacing corn in different proportions on rumen fermentation of Angus bulls.

Items	PBP to Replace Corn in Different Proportions ^1^				SEM	*p*-Value		
	0% PBP	12.84% PBP	25.65% PBP	38.44% PBP		Treatment	Linear	Quadratic
pH	6.63	6.12	6.12	6.18	0.077	0.076	0.078	0.061
NH_3_-N, mg/dL ^2^	7.18	7.38	6.97	7.03	0.305	0.967	0.760	0.919
Acetate, mmol/L	58.21	58.83	56.30	59.76	3.856	0.992	0.957	0.875
Propionate, mmol/L	19.19	21.64	19.72	27.76	2.002	0.465	0.219	0.531
Isobutyrate, mmol/L	0.86	0.61	0.74	0.58	0.072	0.561	0.341	0.714
Butyrate, mmol/L	9.32	12.11	12.35	10.30	0.971	0.687	0.804	0.242
Isovalerate, mmol/L	1.22	0.92	0.93	1.01	0.091	0.692	0.428	0.536
Valerate, mmol/L	1.39	1.43	1.06	1.26	0.117	0.714	0.506	0.797
TVFA ^3^, mmol/L	90.19	96.00	91.20	100.46	6.463	0.952	0.696	0.919
A:P ratio ^4^	3.08	2.88	2.91	2.52	0.092	0.225	0.063	0.629

Abbreviations: ^1^ 0% PBP, control group; 12.84% PBP, 25% replacement group; 25.65% PBP, 50% replacement group; 38.44% PBP, 75% replacement group. ^2^ NH_3_-N, ammonia nitrogen. ^3^ TVFA, total volatile fatty acids. ^4^ A:P ratio, acetate–propionate ratio.

## Data Availability

The original contributions presented in the study are included in the article/Appendix A; further inquiries can be directed to the corresponding author.

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
