# Peer review of "From Food Waste to Sustainable Agriculture: Nutritive Value of Potato By-Product in Total Mixed Ration for Angus Bulls"

_foods, 2024, doi:10.3390/foods13172771_

Round 1

Reviewer 1 Report

Comments and Suggestions for Authors

The study addresses the critical issue of food waste by exploring the use of potato by-products (PBP) in animal feed. This approach contributes to reducing food waste and promotes a circular economy in agriculture. 

The study provides a valuable use for potato by-products, enhancing their economic value and providing an additional income stream for potato processors. Using PBP as a feed ingredient can be more cost-effective than traditional grain-based feeds, potentially lowering the feed costs for farmers. Despite the decrease in apparent digestibility and body weight with higher PBP levels, the study concludes that using PBP does not negatively impact rumen fermentation. This suggests that PBP can be a viable alternative in practical feed formulations.

Limitations of the study:

The significant decrease in body weight with increased PBP levels is a notable limitation. This could be a concern for producers focused on maximizing growth rates and meat production.

The linear decrease in dry matter and starch apparent digestibility with higher PBP levels indicates that PBP might not be as efficiently utilized as traditional grains. This could limit its effectiveness as a feed ingredient at higher inclusion rates.

The changes in rumen microbiota composition, particularly the increase in certain bacteria like Succiniclasticum and Ruminococcus, might have implications for animal health and digestion that need further investigation.

The study focuses on Angus bulls, and the results may not be directly applicable to other breeds or types of livestock. Further research is needed to generalize the findings across different animal species and breeds.

If the study was conducted over a relatively short period, it might not capture the long-term effects of PBP inclusion in the diet. Long-term studies are necessary to understand the sustained impacts on animal health and production.

The nutrient composition of potato by-products can vary significantly depending on the source and processing methods. This variability can affect the consistency and predictability of using PBP in feed formulations.

Conclusion: the paper can offer more robust and applicable findings, making a stronger contribution to the field of sustainable agriculture and animal feed research.

Add the objective of the study at the end of the introduction

3.1. Subsection ?? remove this word for the text

Reviewer 2 Report

Comments and Suggestions for Authors

The manuscript “From food waste to sustainable agriculture: Nutritive value of potato by-product in total mixed ration for Angus bulls” deals with compelling case for the potential use of raw potato fries, a type of potato by-product (PBP), as a partial replacement for grain in the total mixed ration (TMR) of Angus bulls. Before the final decision is made the manuscript need to be revised.

Comments

·       The study mentions that 60 Angus bulls were "randomly assigned" to treatments, but more details on the randomization process (e.g., stratified, block randomization) should be provided.

·       Ln no. 47-53: Please rewrite the content.

·       What were the specific nutritional compositions of the PBP compared to the corn, and how might these differences have influenced the study’s outcomes?

·       The use of a control group (0% PBP) is appropriate, but additional groups with intermediate PBP levels (e.g., 10% or 20%) could provide more granular insights into the effects of PBP replacement.

·       It seems the authors have used Graph Pad Prism for analysis of the graph but have not acknowledged in the material and method section. Kindly do the needful.

·       How was the body weight of the bulls measured, and were there controls for potential variations in weighing conditions?

·       More comprehensive information on the nutritional composition of the TMR (beyond just the levels of PBP) is necessary to understand how other dietary components might have influenced the outcomes. This can be provided in the material and method section.

·       Were there any health or welfare concerns observed in the bulls, particularly in the groups with higher PBP levels?

·       The study lacks a detailed analysis of the nutritional content of PBP, such as the levels of resistant starch, glycemic index, and bioactive compounds. Author can atleast discuss this aspect in the discussion section with related to the parameter studied.

·       The changes in the abundance of specific bacteria like Succiniclasticum and Ruminococcus are noted, but the functional roles of these bacteria in rumen fermentation are not thoroughly explored. Please comment on this aspect.

·       There is no mention of correlation analysis between body weight, digestibility, and microbiota changes, which could provide deeper insights into the interrelationships among these variables.

·       Were the sensory attributes of the beef (e.g., flavor, tenderness) from bulls fed PBP diets assessed, and how did they compare to those fed a traditional corn-based diet?

·       The decrease in dry matter and starch apparent digestibility with increased PBP substitution suggests that the nutritional balance of the diet may have been compromised. This could potentially result in suboptimal energy availability for the cattle, which might explain the reduction in growth performance. The study could benefit from a more detailed analysis of the nutrient composition of the TMR at each substitution level to understand the impact on cattle nutrition better.

Comments on the Quality of English Language

Minor revision is required for language correction

Round 2

Reviewer 2 Report

Comments and Suggestions for Authors

The authors made significant changes in the manuscript.